# Vibrational couplings and energy transfer pathways of water's bending mode

Chun-Chieh Yu[1,6], Kuo-Yang Chiang [1,6], Masanari Okuno[2✉], Takakazu Seki [1], Tatsuhiko Ohto[3], Xiaoqing Yu[1], Vitaly Korepanov [4,5], Hiro-o Hamaguchi[4], Mischa Bonn[1], Johannes Hunger [1✉] & Yuki Nagata [1✉]

Coupling between vibrational modes is essential for energy transfer and dissipation in condensed matter. For water, different O-H stretch modes are known to be very strongly coupled both within and between water molecules, leading to ultrafast dissipation and delocalization of vibrational energy. In contrast, the information on the vibrational coupling of the H-O-H bending mode of water is lacking, even though the bending mode is an essential intermediate for the energy relaxation pathway from the stretch mode to the heat bath. By combining static and femtosecond infrared, Raman, and hyper-Raman spectroscopies for isotopically diluted water with ab initio molecular dynamics simulations, we find the vibrational coupling of the bending mode differs significantly from the stretch mode: the intramode intermolecular coupling of the bending mode is very weak, in stark contrast to the stretch mode. Our results elucidate the vibrational energy transfer pathways of water. Specifically, the librational motion is essential for the vibrational energy relaxation and orientational dynamics of H-O-H bending mode.

[1] Max Planck Institute for Polymer Research, Ackermannweg 10, 55128 Mainz, Germany. [2] Department of Basic Science, Graduate School of Arts and Sciences, The University of Tokyo, 3-8-1 Komaba, Meguro, Tokyo 153-8902, Japan. [3] Graduate School of Engineering Science, Osaka University, 1-3 Machikaneyama, Toyonaka, Osaka 560-8531, Japan. [4] Department of Applied Chemistry and Institute of Molecular Science, National Chiao Tung University, 1001 Ta-Hsueh Road, Hsinchu 30010, Taiwan. [5] Institute of Microelectronics Technology and High Purity Materials, RAS, Chernogolovka, Russia 142432. [6]These authors contributed equally: Chun-Chieh Yu, Kuo-Yang Chiang. ✉email: cmokuno@mail.ecc.u-tokyo.ac.jp; hunger@mpip-mainz.mpg.de; nagata@mpip-mainz.mpg.de

Liquid water is important as a solvent, a solute, a reactant, and a catalyst, in the environment and many technological applications, but also in biology, driving protein folding and structuring nucleic acids. Many of the unique properties of water originate from the hydrogen bond (H-bond) network of water, which results from the strong intermolecular interactions between water molecules. For instance, the water H-bond network allows for rapid delocalization and dissipation of excess energy following chemical reactions, thereby making reactions irreversible. Insights into the flow of excess vibrational energy in water have therefore been deemed essential for understanding this anomalous liquid[1,2].

The vibrational modes of liquid $H_2O$ consist of the O–H stretch mode, the H–O–H bending mode, the three librational modes, and collective modes below $100\,cm^{-1}$. The $3000–4000\,cm^{-1}$ O–H stretch mode has been most intensely studied, owing to its strong response[3]. Both experimental and simulation studies of the O–H stretch mode have demonstrated different parallel pathways of vibrational energy transfer: within one water molecule, and between different water molecules[1,4–7]. Within one molecule, coupling can occur between the stretch modes of the two O–H groups[8], and strong intermolecular coupling of stretch modes makes its excitation quasi-instantaneously delocalized across several molecules in liquid water. The stretch modes are further coupled to the bending mode and its overtone via anharmonic interactions[9–17]. As such, the bend vibration in liquid water is an important intermediate in the vibrational relaxation of the excited O–H stretch vibration to the hot ground state[18–21].

Remarkably, although the stretching mode has been intensely investigated, information on energy transfer pathways from the H–O–H bending mode is scarce. With approximately half the energy of the stretch mode, the bending mode likely provides an efficient pathway for energy delocalization and dissipation. The few experimental and theoretical studies of the bending mode reported to date have revealed that the delocalized character of H–O–H bending mode affects the vibrational dynamics[16,22–24], but the impact of the delocalization on the dynamics is still under debate. For example, the impact of intermolecular bending–bending vibrational coupling (Fig. 1a)[16,22,23,25] as well

as vibrational coupling of the bending and librational modes (Fig. 1b, c)[12–14,23,26] on energy relaxation have been separately investigated, while there is no systematic study for these mechanisms. Furthermore, the impact of the bending mode-librational mode mixing (Fig. 1d) on the dynamics has not been identified, though recent studies imply the importance of the mode mixing beyond the normal mode description[15,16]. This missing understanding of water's bending mode vibrational dynamics precludes a unified view of vibrational energy exchange and relaxation of water vibrations.

In this paper, we explore the energy dynamics of the bend vibration by directly comparing line broadening, vibrational energy relaxation dynamics, and randomization of the transition dipole orientation for the bending mode in $H_2O$-$D_2O$ mixtures. Lowering the $H_2O$ concentration allows suppressing the effect of intermolecular coupling, while keeping the local environment nearly unchanged. Our results show that the intermolecular coupling of the bend vibration is relatively weak in contrast to the stretch vibration. In line with earlier reports[1,3,27], we find intermolecular vibrational energy relaxation to occur on a (sub-)picosecond timescale. Yet, our results suggest that intermolecular bend-to-bend energy transfer is much slower (~1 ps) than the intermolecular stretch-to-stretch transfer (~0.1 ps). Randomization of the orientations for the H–O–H bending mode transition dipole moment occurs much faster, which is attributed to rotational contributions (e.g., librational mode) to the bending mode band.

## Results

### Variation of FTIR, Raman, and hyper-Raman spectra of water bending mode due to vibrational couplings.

The linewidth of a vibrational band is the most obvious reporter of vibrational couplings. By varying the H/D isotopic ratio in water, one can modulate the vibrational couplings of water molecules systematically, while keeping the local environment virtually unaffected[6]. Using this approach, the delocalization of the O–H stretch mode[7,10,28,29] has been demonstrated. This delocalization is also responsible for a significant broadening of the O–H stretch

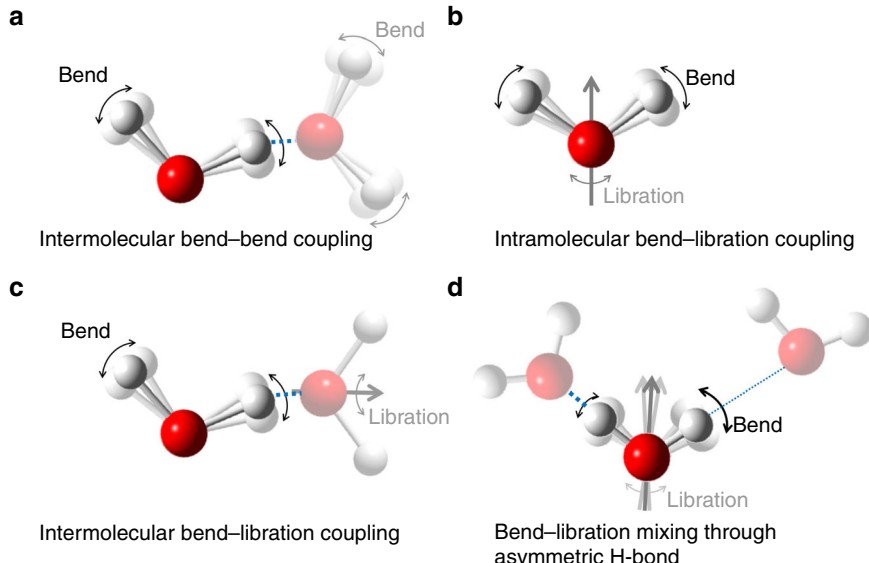

**Fig. 1 Types of the vibrational couplings of the H-O-H bending mode. a** Intermolecular bend-bend coupling of different water molecules. **b** Intramolecular bend-libration coupling within a single water molecule. **c** Intermolecular bend-libration coupling of different water molecules. **d** Bend-libration mixing induced by the asymmetric hydrogen-bond (H-bond) strengths. Hydrogen and Oxygen atoms are represented by white and red spheres, respectively. The gray arrows represent the H-O-H angular bisector direction. Strong (weak) hydrogen-bonds are represented by thick (thin) broken blue lines.

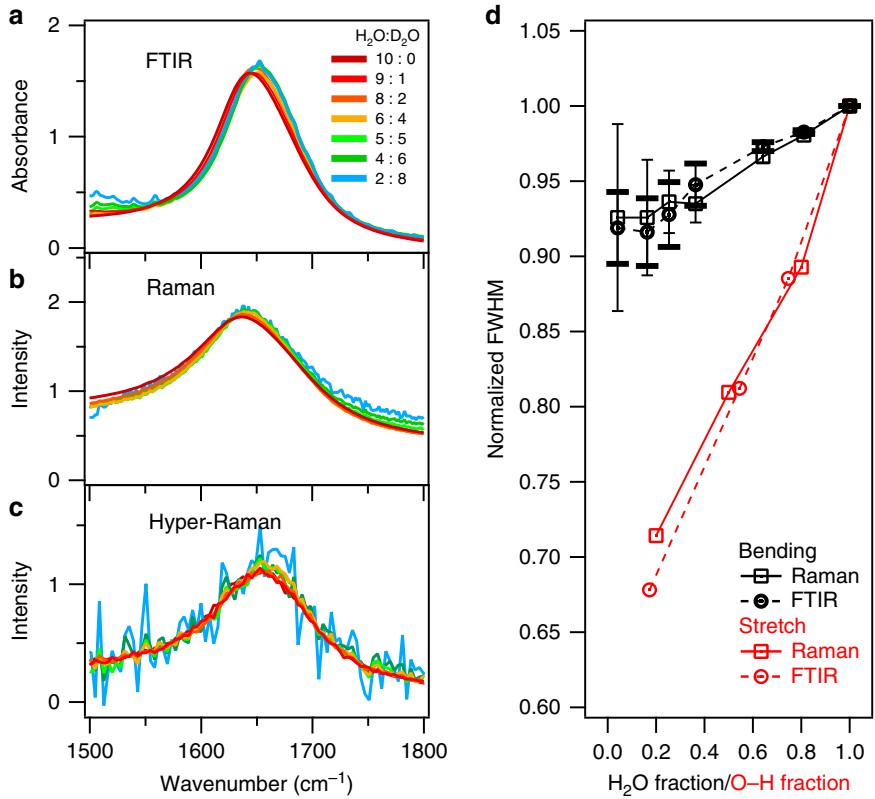

**Fig. 2 The extracted H–O–H bending mode contributions. a** IR, **b** Raman, and **c** hyper-Raman spectra of various $H_2O$-$D_2O$ mixtures. **d** Comparison of the full widths at half maximum (FWHM) for the H–O–H bending mode contributions and the O–H stretch mode spectra. The O–H stretch IR and Raman data are extracted from refs. [49,50], respectively. The error bars correspond to the uncertainty of the FWHMs when using the sample with the different concentration for obtaining the H–O–D bending spectrum for the background subtraction (see Supplementary Note 7).

band for pure $H_2O$. Using isotopic dilution, the vibrational coupling of the bending mode of water can also be modulated. The contributions of background-corrected H–O–H bending mode in FTIR, Raman, and hyper-Raman spectra of $H_2O$-$D_2O$ mixtures are shown in Fig. 2a–c, respectively. The methodology to subtract the background as well as the HDO and $D_2O$ contributions from the measured spectra is detailed in the Supplementary Note 6. The IR, Raman, and hyper-Raman responses for the H–O–H bending mode in neat $H_2O$ peak at 1644, 1637, and 1656 $cm^{-1}$, respectively. The variation in the peak frequency between the IR, Raman, and hyper-Raman spectra reflects the different frequency-dependence of the transition dipole moment, polarizability, and hyper-polarizability – the so-called non-coincidence effect – and also signifies coupling[30,31]. While the peak frequencies are different for the three techniques, the blue-shift of the resonance upon isotopic dilution is very similar. The similarity suggests that the underlying vibrational couplings in water have a common origin[32,33]. The insensitivity of the magnitude of the shift to the experimental details is consistent with the notion that intermolecular bend–bend coupling is not of dipolar nature but has mechanic or electronic origins[32]. Accordingly, the same blue-shift can be observed in the aqueous salt solutions (see Supplementary Figs. 10 and 11).

The variations of the full widths at half maximum (FWHM) are summarized in Fig. 2d. The FWHMs of the H–O–H bending mode decrease in both the IR and Raman spectra upon the addition of $D_2O$. This means that the vibrational coupling of the bending mode not only shifts the peak frequency but also broadens the linewidths. On the other hand, the variation of the FWHM for the bending mode is much smaller than that of the O–H stretch mode which narrows dramatically upon addition of

$D_2O$. The small FWHM variation of the bending mode, compared to the stretch mode, can be attributed to weaker intermolecular intramode transition dipole interaction resulting from the smaller transition dipole moment for the bending mode, relative to the O–H stretch mode[34]. Furthermore, the degree of narrowing is much smaller than expected from previous theoretical predictions[32]. These observations show that the vibrational coupling effects on the peak width and position are modest, yet clearly present in the static spectra.

**Energy relaxation from water bending mode to lower frequency mode determined by pump-probe IR measurement.** To disentangle these vibrational couplings of the H–O–H bending mode, we carried out time-resolved IR measurement by pumping and probing the H–O–H bending mode in neat $H_2O$ and isotopically diluted water. In these experiments, an intense pump laser pulse excites the bending mode from the vibrational ground state to the first vibrationally excited state, and transient spectral modulations are probed as a function of delay time, $t$. The excitation reduces absorption at the fundamental frequency of the bending mode (ground state bleach) and increases absorption (excited state absorption) at red-shifted frequencies. We measured the transient absorption spectra with the polarization of the probe light both perpendicular ($\Delta\alpha_\perp(\omega, t)$) and parallel ($\Delta\alpha_\parallel(\omega, t)$) directions to that of the excitation pulse.

First, we discuss the vibrational energy relaxation (intramode energy transfer) of the H–O–H bending mode. In neat $H_2O$, the vibrational energy transfer from the bending mode to librational mode occurs within a single water molecule (intramolecularly, Fig. 1b) and through energy relaxation from the bending mode of

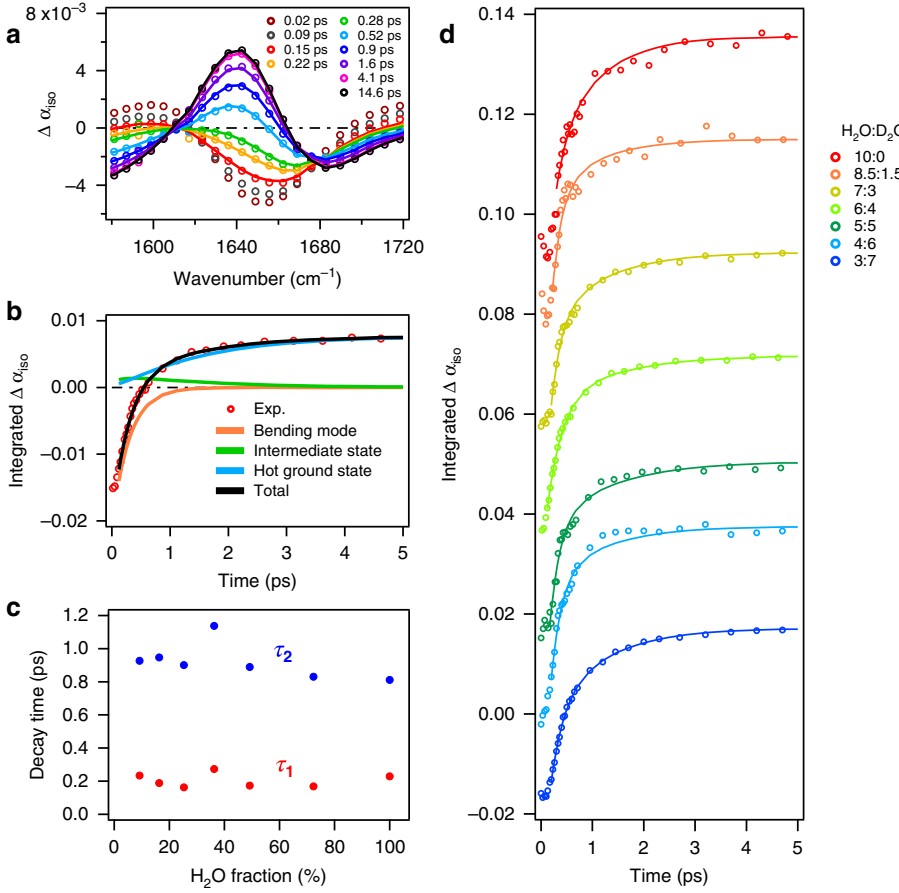

**Fig. 3 The isotropic transient absorption of various isotopic compositions. a** Transient absorption spectra of water bending mode for $H_2O:D_2O = 6:4$. The four-state model fit the data starting at 0.15 ps. **b** Time evolution of the transient absorption spectra integrated at $1649 \, cm^{-1} < \omega < 1662 \, cm^{-1}$. The orange, green, and blue lines represent the contributions of the excited state, intermediate state, and hot ground state, respectively, to the fit. **c** Isotropic relaxation times as a function of isotopic composition. The relaxation time of the excited state to intermediate state is $\tau_1$ (red symbol) and the relaxation time of the intermediate state is $\tau_2$ (blue symbol). **d** Isotropic transient signals for various $H_2O$-$D_2O$ mixtures. The signals are normalized at $t = 0$. Traces are offset by increments of 0.02. Open circles in panels **a**, **b** and **d** represent the experimental data, while the solid lines show fits with the kinetic model.

one water molecule to the librational mode of an adjacent water molecule (intermolecularly, Fig. 1c)[12–14,23,26]. Through isotopic dilution of water, the intermolecular intermode transfer from the bending mode of the $H_2O$ molecule to the librational mode of the surrounding water molecules is altered, because the isotopic dilution shifts the librational mode frequency of $H_2O$ to that of $D_2O$ and HDO (see also Supplementary Figs. 2 and 3), changing the vibrational coupling between the bending mode and librational mode.

Figure 3a shows the isotropic component of the transient absorption response, $\Delta\alpha_{iso}(\omega, t) = [\Delta\alpha_{||}(\omega, t) + 2\Delta\alpha_{\perp}(\omega, t)]/3$, which solely represents energy relaxation and is free from rotational contributions, of the H–O–H bending mode for the $H_2O:D_2O = 6:4$ sample. With increasing delay time, the excited state population relaxes to the ground state, and the dissipated energy populates low frequency modes, which results in persistent spectral modulations at long delay times. To quantify the vibrational dynamics, we used a four-state kinetic model similar to previous studies[12,14], in which the four states represent, respectively, ground state, bending mode excited state, intermediate state (e.g. including the librational mode excited state), and hot ground state. Figure 3b shows integrated ($1649 \, cm^{-1} < \omega < 1662 \, cm^{-1}$) $\Delta\alpha_{iso}$ signals as a function of delay time together with the overall fit and the contributions of each state of the model (for details on the fit model, see Supplementary Note 8);

the fit manifests that the four-level model describes the experimental data very well (Fig. 3a, b). Figure 3d displays the integrated $\Delta\alpha_{iso}(\omega, t)$ for various concentrations of $H_2O$-$D_2O$ mixtures. The time traces are similar for all samples. The relaxation time from the excited state bending mode to the intermediate state ($\tau_1$) is ~200 fs (Fig. 3c), consistent with previous reports[14,16]. Within error, $\tau_1$ is insensitive to the isotopic composition. The insensitivity of $\tau_1$ to $H_2O$ concentration reveals that the energy relaxation process is unaffected by the isotopic composition of the water molecules surrounding the excited chromophore. This suggests that intramolecular (Fig. 1b), but not intermolecular (Fig. 1c), bend-libration coupling dictates the vibrational energy relaxation of the bending mode.

**Intermolecular bend-to-bend energy transfer measured by polarization-dependent pump-probe IR measurement.** Figure 4a shows the time traces of the anisotropy decay for different isotopic substitutions. The decay of the excitation anisotropy of the H–O–H bending mode, i.e., the orientational memory of the anisotropic excitation due to polarized light, has provided detailed insight into the coupling of the H–O–H bending chromophores[7,35]. This anisotropy decay is defined as;

$$R(t) = \int_{\omega_1}^{\omega_2} \frac{\Delta\alpha'_{||}(\omega, t) - \Delta\alpha'_{\perp}(\omega, t)}{\Delta\alpha'_{||}(\omega, t) + 2\Delta\alpha'_{\perp}(\omega, t)} d\omega \qquad (1)$$

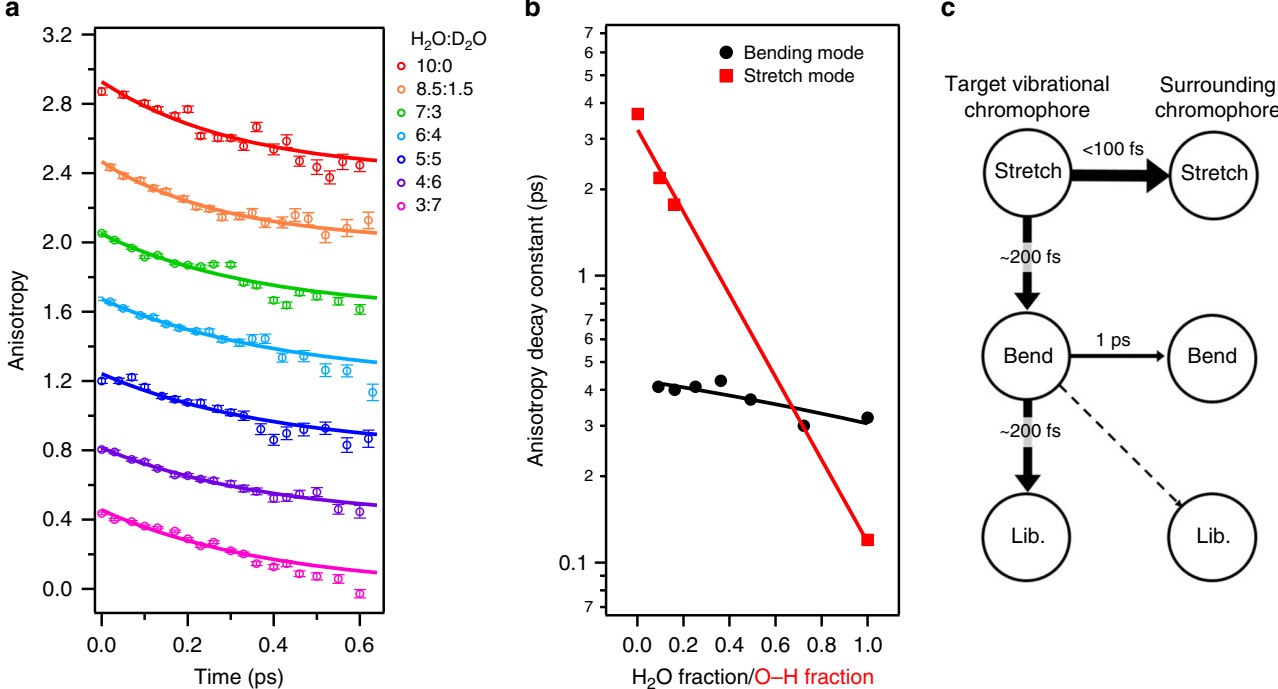

**Fig. 4 The anisotropy decay of various isotopic composition and schematic for the vibrational energy relaxation of water molecules. a** Anisotropy decay for various $H_2O$-$D_2O$ mixtures. To obtain the excitation anisotropy, the signals arising from the hot ground state were subtracted from the raw data. Traces are offset by increments of 0.4. Symbols show experimental data and solid lines show fits using a single exponential decay. **b** Comparison of the anisotropy decay time for the bending mode and the stretch mode as a function of $H_2O$/O–H fraction. The $H_2O$/O–H fraction represent the mole fraction of $H_2O$ molecules and of O–H stretch groups for various $H_2O$-$D_2O$ mixtures, respectively. The stretch mode decay times were obtained by fitting an exponential decay to the data reported in ref. [7]. The lines serve to guide the eye. **c** Schematic representation for the vibrational energy relaxation of H–O–H molecules in neat water. The stretch mode timescales were taken from refs. [15,27,51,52]. The target vibrational chromophore is the initial excited H–O–H molecule. The surrounding chromophores are the water molecules surrounding the excited molecule.

where $\Delta\alpha'_\perp(\omega, t)$ and $\Delta\alpha'_\parallel(\omega, t)$ represent the transient spectra after the effects of the spectral contribution of the hot ground state (i.e., the spectral signature at long $t$, in Fig. 3b) were subtracted from $\Delta\alpha_\perp(\omega, t)$ and $\Delta\alpha_\parallel(\omega, t)$, respectively. We set $\omega_1 = 1649 \, \mathrm{cm}^{-1}$ and $\omega_2 = 1662 \, \mathrm{cm}^{-1}$ in Eq. (1), similar to a previous study[16].

In contrast to the isotropic transient signals, the excitation anisotropy provides insight into the randomization of the transition dipole orientation of the excited chromophore. Such orientational randomization for neat $H_2O$ can arise from three mechanisms; firstly, the rotational motion of water, secondly the loss of the orientational information through exciton hopping to a neighboring water molecule with a differently aligned transition dipole (facilitated by intermolecular bend-to-bend coupling, Fig. 1a), and thirdly the orientational fluctuations caused by bend-libration mixing (Fig. 1d), which all result in a decay of the excitation anisotropy. In isotopically diluted water, the contribution due to intermolecular bend-to-bend energy transfer is suppressed, because of the frequency mismatch of the H–O–H bending mode and H–O–D/D–O–D bending modes. As such, in analogy to the O–H stretching band[7,35], one can quantify the intermolecular bend-to-bend energy transfer from the anisotropy decay of different isotopic composition.

The time evolution of the anisotropy is displayed in Fig. 4a. The anisotropy of $H_2O$ molecules in neat $H_2O$ decays with a timescale of ~300 fs. This decay is somewhat slower than that previously reported for neat $H_2O$[16,22], which can likely be attributed to the different procedure to subtract the contributions of the hot ground state from the data (see the Supplementary Notes 8 and 9). Consistent with these reports, and despite the

short vibrational relaxation time limiting the accessible time window for determining the excitation anisotropy to $t < 600$ fs, we find the anisotropy to decay faster than what would be expected based on rotation of water (i.e., the H–O–H angular bisector orientational decay of 1.9 ps[36]). The anisotropy decay times, as obtained from a fit of an exponential decay to the data, slows down from $310 \pm 20$ fs to $440 \pm 20$ fs upon addition of $D_2O$. This is in stark contrast with the O–H stretch mode, which shows >10 times faster anisotropy decay in neat $H_2O$ than in isotopically diluted water due to the stretch-stretch modes coupling/energy transfer (see Fig. 4b)[1,7]. The weaker variation of the anisotropy with isotopic composition may be rationalized by the smaller transition dipole moment of the bending mode of water. Assuming that the variation of the anisotropy decay can be fully ascribed to intermolecular bend-to-bend energy transfer, we estimated the time constant of bend-to-bend energy transfer in neat water from the difference in the fitted decay rates at $H_2O$ fractions of 0 and 1. This provides the time constant of $((310 \, \mathrm{fs})^{-1} - (440 \, \mathrm{fs})^{-1})^{-1} = 1.05 \pm 0.26$ ps. As such, the intermolecular intramode transfer is rather slow, in stark contrast to the O–H stretch mode, exhibiting strong intermolecular intramode vibrational couplings. In turn, the vibrational energy transfer pathway from $H_2O$ bending modes seems to be governed by the energy relaxation to the librational motion. This is summarized in Fig. 4c.

## Discussion
The fact that the anisotropy of the bending excitation decays on a 440-fs time scale, which is much faster than the reorientation time of a water molecule (~1.9 ps[36]) strongly suggests that

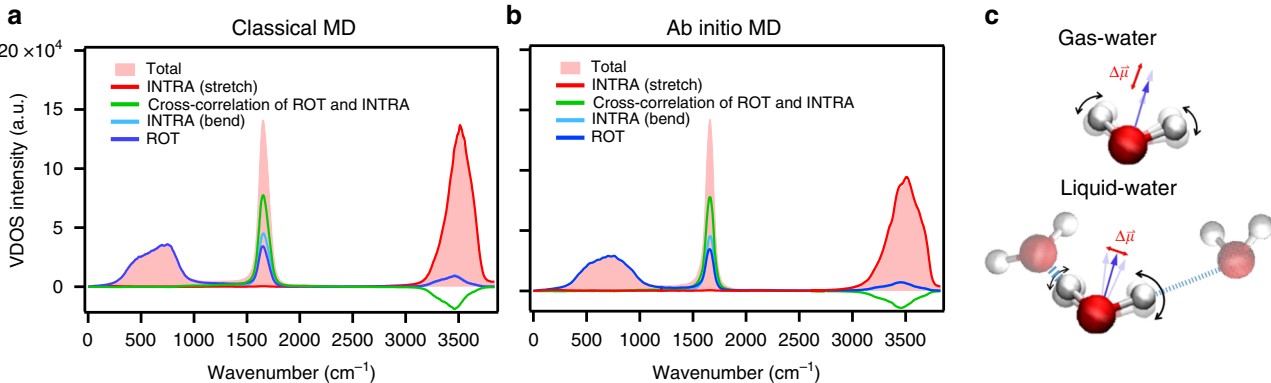

**Fig. 5 Decomposed vibrational density of states (VDOS) spectra of neat H₂O. a** Classical force field MD with the POLI2VS model[53] and **b** AIMD simulation at the revPBE0-D3(0) level of theory. The frequency of the POLI2VS model is scaled with a factor of 0.96 to account for nuclear quantum effects. The frequencies of the bending and stretch modes for the revPBE0-D3(0) model are corrected via the procedure described in ref. [54]. The red region illustrates the total VDOS spectrum of neat H₂O. The differently colored lines represent the contribution of different motions. **c** Schematics of water molecules in the gas and liquid phase. Strong (weak) hydrogen-bonds are represented by thick (thin) broken lines. The blue arrows represent the direction of the dipole moment of the highlighted water molecule, while the red lines represent the direction of the change in the dipole moment (transition dipole moment, $\Delta\vec{\mu}$).

vibrational modes other than the pure bending motion contribute to the observed band at ~1650 cm⁻¹ [16]. If the 1650 cm⁻¹ band contains vibrational contributions other than a pure bending motion of water's protons, the detected anisotropy decay does not solely arise from the pure rotational motion of the water molecule, rationalizing the anisotropy decay of the bending mode to be faster than the rotation times as obtained from, e.g., IR pump-probe experiments on isotopically diluted HOD molecules and NMR experiments[37,38].

To elucidate the exact origin of the vibrational response at 1500–1600 cm⁻¹, we computed the vibrational density of states (VDOS) spectra from the velocity autocorrelation as obtained from ab initio molecular dynamics (AIMD) simulations at the revPBE0-D3(0) level of theory as well as from classical force field simulations with the POLI2VS model. We decomposed the vibrational response based on the decomposition of the velocity into the components that arise from the rotational motion (ROT) of an entire water molecule and the intramolecular motion (INTRA). We further decomposed the INTRA motion into the O–H stretching motion that goes along with a variation of the O–H distances and into bending motion contributions, which mainly involves the variation of the H–H distance. The subsequent decomposed VDOS spectra are displayed in Fig. 5a, b (for computational details, see Supplementary Note 5).

The simulated VDOS spectra show that the 1650 cm⁻¹ feature is not only governed by the intramolecular vibrational motion but also contains contributions from the rotational motion of water molecules. Interestingly and perhaps surprisingly, the pure bending motion, which is not accompanied by a simultaneous rotational motion of water, only accounts for 30% of the 1650 cm⁻¹ VDOS feature. The largest contribution arises from the cross-correlation terms of the bending motion and rotational motion through the mixing of the bending mode and librational mode (see Fig. 1d). As such, the simulations confirm the strong coupling of the bending mode to water's librations, giving rise to the 1650 cm⁻¹ band. The marked rotational contributions to the 1650 cm⁻¹ peak can be rationalized by the notion that – in contrast to water in the gas phase – in liquid water the heterogeneity of hydrogen-bond strengths around a water molecule makes the bending mode potential asymmetric with respect to the bisector (Fig. 5c)[39,40]. The vibrational coordinate of the 1650 cm⁻¹ mode contains both rotational motions of the water molecule and a modulation of the H–H distance. In contrast to the band at 1650 cm⁻¹, the band at

3200–3600 cm⁻¹ can be primarily explained by a proton motion along the O–H axis, that is, the stretching vibration (Fig. 5a, b).

Together, the rotational contributions to the 1650 cm⁻¹ mode make the orientation of its transition dipole to differ from the H–O–H angular bisector (Fig. 5c), and the anisotropy decay of the pump-probe experiments probes the motion of an axis different from the H–O–H angular bisector directions. Since the orientation of the transition dipole moment varies with the hydrogen bond strength of water as schematically depicted in the bottom panel of Fig. 5c, the acceleration of the anisotropy decay with respect to the pure H–O–H bisector orientational motion can be linked to the hydrogen bond dynamics. In fact, the acceleration can be estimated to $((440\ \text{fs})^{-1} - (1.9\ \text{ps})^{-1})^{-1} = 570\ \text{fs}$. The 570-fs time scale agrees with the time constant of the hydrogen bond dynamics of 0.5–1.0 ps, as characterized by the spectral diffusion for the O–H stretch of dilute HOD in D₂O[28,41–43]. As such, large bending-libration mode mixing rationalizes the observation that the excitation anisotropy for the 1650 cm⁻¹ band (~440 fs) of isotopically diluted water as measured in the pump-probe IR measurement decays much faster than the orientation of the bisector as measured using NMR (~1.9 ps)[36]. This large bending-librational mode mixing causes the 10% line broadening of the static bending mode spectra (see Fig. 2). In contrast, the weak mode mixing for the stretch mode of isotopically diluted water is consistent with the observation that the anisotropy decay for the of the O–H stretch mode at ~3400 cm⁻¹ for isotopically diluted water (≈3–4 ps)[7,41,44] is comparable to the orientational correlation time as detected with NMR (decay ≈2 ps)[45–48].

To summarize, we quantified the vibrational coupling of the water bending mode employing static and time-resolved vibrational spectroscopic techniques. The IR, Raman, and hyper-Raman measurements of water bending mode show that the vibrational coupling of the bending mode blue-shifts the peak position by ~8 cm⁻¹ and reduces the linewidth by 10%, upon isotopic dilution. The time-resolved IR measurements reveal that the intermolecular bend-to-bend (intramode) coupling is much weaker than the intermolecular stretch-to-stretch coupling. The intramolecular bend-to-libration energy transfer (≈200 fs) takes place much faster than the intermolecular bend-to-bend coupling (≈1 ps). As such, the vibrational energy of bending mode is mainly released not to the bending modes of surrounding water molecules but to the librational mode, in stark contrast to the stretch mode. AIMD simulations confirm strong bend-libration coupling, which can explain the faster decay of the excitation anisotropy of the bending mode (≈0.4 ps), much

faster than for the stretch mode ($\approx$2–4 ps). The experimental and simulation results demonstrate that the librational mode plays a key role in the energy transfer pathway and orientational dynamic of the bending mode band.

## Methods

**FTIR, Raman, and hyper-Raman experiment.** For the FTIR experiment, FTIR spectra were recorded with a Bruker VERTEX 70 FTIR spectrometer in transmission. The samples were held between two CaF$_2$ windows separated by a 15-µm-thick Teflon spacer. The spectrometer was purged with N$_2$ and spectra were recorded at 298 K. For the Raman experiment, the excitation was made with a 532-nm laser (Millennia eV, Spectra-Physics). The signal was dispersed in a spectrometer (SR303i-B, Andor) and detected by a CCD camera (DU420A-BVF, Andor). We measured the spectra in the parallel polarization configuration. For the hyper-Raman experiment, we used a picosecond laser (Cepheus 1002, Photon Energy, 1064 nm, ~15 ps, 150 kHz). The output was frequency-doubled (532 nm) and then focused into the sample. The signal was dispersed in a spectrometer (iHR320, Horiba) and detected by a CCD camera (DU420A-BVF, Andor). The details can be found in the Supplementary Notes 1–3.

**Pump-probe IR experiment.** The measurements were performed on a femtosecond Ti: Sapphire amplified laser system (Coherent Astrella, ~800 nm, ~35 fs, 1 kHz) with 6.8 W output power. In all, 2.8 W of the output was used to pump an optical parametric amplifier (TOPAS, light conversion) with a non-collinear DFG stage to generate broadband IR pulses (centered at 1600 cm$^{-1}$, 100 fs duration, 17 µJ pulse energy, and 300 cm$^{-1}$ full width at half maximum (FWHM)). The IR pulses were split into probe (~2.5%), reference (~2.5%), and pump pulses (~95%). The polarization of the pump-pulse was set at 45° with respect to the probe pulse polarization using a half-wave plate. The sample was placed between two CaF$_2$ windows separated by a 15-µm-thick Teflon spacer. A wire grid polarizer mounted in a rotating stage allowed us to select the parallel and perpendicular polarization components of the probe beam, relative to the pump polarization. Both the probe and the reference pulses were dispersed with a spectrometer onto a liquid-nitrogen-cooled mercury-cadmium telluride detector. The details can be found in the Supplementary Note 4.

**MD Simulation.** We used the AIMD trajectories at the revPBE0-D3(0) level of theory as well as the classical force field molecular dynamics trajectories with the POLI2VS model. The simulation cell contained 64 H$_2$O in the AIMD, while the POLI2VS simulation cell contained 500 H$_2$O molecules in the POLI2VS simulation. The target temperature was set to 300 K. The VDOS spectra were calculated for the dipole moment directions via;

$$\text{VDOS}(\omega) = \int_0^T \cos(\omega t)\cos^2\left(\frac{\pi t}{2T}\right)\langle \Sigma_i \mathbf{v}_i(t) \cdot \mathbf{v}_i(0)\rangle \mathrm{d}t, \tag{2}$$

$$\mathbf{v}_i(t) = \frac{\left(\mathbf{v}_{i,\text{H1}}(t) + \mathbf{v}_{i,\text{H2}}(t)\right)}{2} - \mathbf{v}_{i,\text{O}}(t), \tag{3}$$

where $\mathbf{v}_{i,x}(t)$ denotes the velocity vector of atom $x$ = O, H$_1$, H$_2$ for water molecule $i$. The VDOS spectra were decomposed based on the center of mass velocity $\mathbf{v}_{i,\text{COM}}(t)$, rotational motion velocity $\mathbf{v}_{i,\text{ROT}}(t)$, intramolecular bending motion velocity $\mathbf{v}_{i,\text{INTRA(bend)}}(t)$, and intramolecular stretch motion velocity $\mathbf{v}_{i,\text{INTRA(stretch)}}(t)$. The details can be found in the Supplementary Note 5.

## Data availability
The data that support the findings of this study are available from the corresponding authors upon reasonable request.

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

## Acknowledgements

We acknowledge the financial support from the European Research Council (ERC) under the European Union's Horizon 2020 research and innovation program (grant agreement no 714691), DAAD (grant agreement no 57526761), and from the MaxWater Initiative of the Max Planck Society. We thank to Sho Imoto, Jiyu Xu, and Seonchoel Cha for fruitful discussion.

## Author contributions

Y.N. and J.H. conceived the research idea. C.-C.Y. and K.-Y.C. conducted the IR and pump-probe IR experiments, and M.O. conducted the Raman and hyper-Raman experiments. C.-C.Y., K.-Y.C., and J.H. analyzed the experimental data. Y.N. and T.O. carried out the simulations and analyzed the simulation data. C.-C.Y., K.-Y.C., M.O., T.S., T.O., X.Y., V.K., H.H., M.B., J.H., and Y.N. discussed and interpreted the results. C.-C.Y., K.-Y.C., M.B., J.H., and Y.N. wrote the manuscript.

## Funding

## Competing interests

The authors declare no competing interests.
