## [Peer Review File · Nature Communications]

REVIEWER COMMENTS

Reviewer #1 (Remarks to the Author):

The paper by Yu et al. describes a steady state and time resolved study of the relaxation of the water bending mode. Contrary to what is stated at the bottom of page 3 (though clearly evident from the citations, which seem quite proper) this problem has been studied in some detail for more than ten years, both experimentally and through simulation. The overall conclusions from the earlier work, that relaxation to the libration dominates, are in line with the conclusions reached here. What seems genuinely new in this work are the isotopic dilution experiments, which help to resolve intermolecular from intermolecular pathways. This new data is significant, but isotopic dilution should be mentioned in the title or, at least, the abstract.

The spectral linewidth, the population relaxation and the anisotropy decay are all analysed as a function of dilution. Rather small effects are seen, consistent with a dominant 'intramolecular' pathway. The significance of the separate Raman, hyper-Raman and IR spectra was not clearly explained – what does each measurement bring that is unique? Brief mention was made of the observed non-coincidence effect. This an important effect, which reflects intermolecular coupling, but is not followed up in subsequent discussion. If this is to be neglected, then what is the information in these three spectra?

On page 6 the statement 'the variation of the bending mode's FWHM is much smaller due to its smaller transition dipole' requires additional explanation. Describe the role of the transition dipole?

On Page 8 the statement 'because the isotopic dilution shifts the librational mode frequency,' needs further comment. Why is the libration more shifted than the bend? Why is the librational spectrum not shown (the data are cut at 1100 cm⁻¹)?

Please show and briefly discuss the breakdown of Figure 3a into its assigned parts. What is the intermediate state – a librationaly excited water molecule?

Are the 0.28 ps data missing from figure 3a?

To analyse the anisotropy decay has the Condon approximation been made, and is it appropriate?

The discussion of the anisotropy data seems reasonable, but the 'faster than rotation' assignment needs some more discussion. The single molecule rotation time is 2 ps. Given the ca 200 fs population relaxation could pure rotation have been observed by this method. How is the anisotropy measurement influenced by the intermediate population?

Minor points

P11 4 lines from the bottom, 'and thirdly fluctuations' In what?

P12 4 lines from bottom, there is a misplaced comma.

In summary, the paper presents some new data on an important aspect of water dynamics. The conclusions reached are broadly in line with earlier work. The discussion in some places lack detail (see above). A journal such as J Chem Phys might allow for a more rigorous discussion.

Reviewer #2 (Remarks to the Author):

In this paper, the authors experimentally and through molecular dynamics simulations studied the vibrational coupling of the H-O-H bending mode of liquid water. The vibrational energy relaxation and orientational dynamics

of H-O-H bending mode were also studied. The authors established that the intermolecular coupling of the bending mode is very weak. This fact has been known for a while (see e.g. J. Chem. Phys. 143, 014502 (2015)). Importantly, the authors found that there is a strong coupling of the bending mode and librations which gives

rise to the 1650 cm^{-1} band and demonstrated that the librational mode plays a key role in the energy

transfer pathway and orientational dynamic of the bending mode band. This paper should be of considerable interest to researchers studying condensed-phase water. Publication in Nat. Comm. is recommended. I have only two minor suggestions.

Page 3: "The O-H stretch vibration in liquid water is quasi-instantaneously delocalized

across several water molecules, and is strongly coupled to the bending mode". Considering a substantial difference in vibrational frequencies it is not clear why the authors say that these modes are strongly coupled. The authors should be more specific about what they mean by coupling here.

Page 6. "Furthermore, the degree of narrowing

is much smaller than expected from previous theoretical predictions [29]". Ref. 29 illustrates a very good agreement between theory and experiment, yet the authors of this paper say that the narrowing is smaller. How can these two facts be reconciled? In other words, why the experimental data presented in this paper appear to be different from the experimental data used as a reference in Ref. 29?

Reviewer #3 (Remarks to the Author):

This manuscript describes a combined experimental and theoretical investigation aimed at characterizing the inter and intramolecular couplings and energy dissipation pathways of the water bending mode. Where stretch mode have been extensively studied, the bending mode is understudied despite it being essential to the energy dissipation of water. The study explains the much different inter vs intra-molecular contributions to the mode coupling of the bend vs the stretch and further show that the bending mode is heavily influences by libration modes giving rise to a much different mode character.

This is a very interesting study that is well-carried out and described. It will be interesting to a broad audience and appropriate for Nature Communications.

My only comments pertain to a couple of wording choices that needs to be corrected before publication:

Page 3: "Within one molecule, coupling can occur between the symmetric and antisymmetric O-H stretch modes8,"

The paper from the Tokmakoff Group (Reference 8) show that the mode character is dramatically different between liquid D2O and H2O. In liquid D2O there remain some character of the gas phase symmetric and asymmetric stretch modes but this is not the case for liquid H2O. They authors

statement is thus misleading since this is only partly true for D₂O and not for H₂O. The statement should be reformulated so something like: “Within one molecule, coupling can occur between the two OH stretch modes”

Page 9: “the excited state population relaxes to the ground state, and the dissipated energy gives rise to a temperature increase, which results in persistent spectral modulations at long delay times.”

This is partly semantics, but at the probed timescales the dissipated energy among low-frequency modes cannot be described by a thermal distribution and thus cannot be described as a temperature increase. Only at much longer timescales than is probed in this experiment can this be described as a temperature increase. The statement should be reworded accordingly.

REPLY TO REVIEWER COMMENTS

Reviewer #1 (Remarks to the Author):

The paper by Yu et al. describes a steady state and time resolved study of the relaxation of the water bending mode. Contrary to what is stated at the bottom of page 3 (though clearly evident from the citations, which seem quite proper) this problem has been studied in some detail for more than ten years, both experimentally and through simulation. The overall conclusions from the earlier work, that relaxation to the libration dominates, are in line with the conclusions reached here. What seems genuinely new in this work are the isotopic dilution experiments, which help to resolve intermolecular from intermolecular pathways. This new data is significant, but isotopic dilution should be mentioned in the title or, at least, the abstract.

The spectral linewidth, the population relaxation and the anisotropy decay are all analysed as a function of dilution. Rather small effects are seen, consistent with a dominant 'intramolecular' pathway. The significance of the separate Raman, hyper-Raman and IR spectra was not clearly explained – what does each measurement bring that is unique? Brief mention was made of the observed non-coincidence effect. This an important effect, which reflects intermolecular coupling, but is not followed up in subsequent discussion. If this is to be neglected, then what is the information in these three spectra?

On page 6 the statement 'the variation of the bending mode's FWHM is much smaller due to its smaller transition dipole' requires additional explanation. Describe the role of the transition dipole?

On Page 8 the statement 'because the isotopic dilution shifts the librational mode frequency,' needs further comment. Why is the libration more shifted than the bend? Why is the librational spectrum not shown (the data are cut at 1100 cm⁻¹)?

Please show and briefly discuss the breakdown of Figure 3a into its assigned parts. What is the intermediate state – a librational excited water molecule?

Are the 0.28 ps data missing from figure 3a?

To analyse the anisotropy decay has the Condon approximation been made, and is it appropriate?

The discussion of the anisotropy data seems reasonable, but the 'faster than rotation' assignment needs some more discussion. The single molecule rotation time is 2 ps. Given the ca 200 fs population relaxation could pure rotation have been observed by this method. How is the anisotropy measurement influenced by the intermediate population?

Minor points

P11 4 lines from the bottom, 'and thirdly fluctuations' In what?

P12 4 lines from bottom, there is a misplaced comma.

In summary, the paper presents some new data on an important aspect of water dynamics. The conclusions reached are broadly in line with earlier work. The discussion in some places lack detail (see above). A journal such as J Chem Phys might allow for a more rigorous discussion.

Authors' Response:

We appreciate the reviewer's insightful comments. In the following, we present a point-by-point to the reviewer's comments.

Comment 1:

This new data is significant, but isotopic dilution should be mentioned in the title or, at least, the abstract.

Authors' Response:

We agree with the reviewer and have included 'isotopic dilution' in the abstract.

Authors' Action:

The revised abstract reads: "By combining static and femtosecond infrared, Raman, and hyper-Raman spectroscopies for isotopically diluted water with *ab initio* molecular dynamics simulations, we find the vibrational coupling of the bending mode differs significantly from the stretch mode"

Comment 2:

The spectral linewidth, the population relaxation and the anisotropy decay are all analysed as a function of dilution. Rather small effects are seen, consistent with a dominant 'intramolecular' pathway. The significance of the separate Raman, hyper-Raman and IR spectra was not clearly explained – what does each measurement bring that is unique? Brief mention was made of the observed non-coincidence effect. This an important effect, which reflects intermolecular coupling, but is not followed up in subsequent discussion. If this is to

be neglected, then what is the information in these three spectra?

Authors' Response:

We thank the reviewer for this valuable suggestion. Indeed, the frequency shifts observed in the different experiments upon isotopic substitution contain important information, which we glossed over in our original submission. In particular, the similarity of the observed blue-shift in the different experiments provides insights into the nature of the coupling. Accordingly, we have revised the discussion of the peak frequencies and the frequency shifts.

Authors' Action:

The revised discussion reads: “The variation in the peak frequency between the IR, Raman, and hyper-Raman spectra reflects the different frequency-dependence of the transition dipole moment, polarizability, and hyper-polarizability – the so-called non-coincidence effect – and also signifies coupling^{29,30}. While the peak frequencies are different for the three techniques, the blue-shift of the resonance upon isotopic dilution is very similar. The similarity suggests that the underlying vibrational couplings in water have a common origin^{31,32}. The insensitivity of the magnitude of the shift to the experimental details is consistent with the notion that coupling of the bending mode is not of dipolar nature but has mechanic or electronic origins³¹. Accordingly, the same blue-shift can be observed in the aqueous salt solutions (see Supplementary Figs. 10 and 11).”

Comment 3:

On page 6 the statement 'the variation of the bending mode's FWHM is much smaller due to its smaller transition dipole' requires additional explanation. Describe the role of the transition dipole?

Authors' Response:

Due to the smaller transition dipole, the line shapes of the bending mode are less affected by the transition dipole-transition dipole interactions as compared to the stretch mode as discussed in J. Phys. Chem. Lett. 2012, 3, 22, 3348–3352.

Authors' Action:

We have added the reference and elaborated the discussion as follows: “The small FWHM variation of the bending mode, compared to the stretch mode, can be attributed to the weaker intermolecular intramode transition dipole interaction resulting from the smaller transition dipole moment for the bending mode, relative to the O-H stretch mode³³. ”

Comment 4:

On Page 8 the statement 'because the isotopic dilution shifts the librational mode frequency,' needs further comment.

Authors' Response:

We apologize for this ambiguous statement. Here, we meant to convey that intermolecular intermode energy transfer from the bending mode of H₂O and intramolecular relaxation from the bending mode occurs. Upon isotopic dilution, only the intermolecular pathway is altered, as the librational resonance frequency of neighboring water molecules is shifted.

Authors' Action:

We have revised the manuscript, which now reads “Through isotopic dilution of water, the intermolecular intermode transfer from the bending mode of the H₂O molecule to the librational mode of the surrounding water molecules is altered, because the isotopic dilution shifts the librational mode frequency of H₂O to that of D₂O and HDO (see also Supplementary Figs. 2 and 3), changing the vibrational coupling between the bending mode and librational mode.”

Comment 5:

Why is the librational spectrum not shown (the data are cut at 1100 cm⁻¹)?

Authors' Response:

For the infrared experiments, the windows (CaF₂) used for the present experiments do not allow recording spectra at librational frequencies. As the main focus of the present study is the bending mode, we have limited data here to bending spectra. The Raman and hyper-Raman experiments, however, contain information on the librational modes of water, which we have included in the revised Supplementary Information.

Authors' Action:

We have added the experimental data (Raman and hyper-Raman spectra) for the librational mode, shown in the Supplementary Information, Supplementary Figures 2 and 3.

Comment 6:

Please show and briefly discuss the breakdown of Figure 3a into its assigned parts.

Authors' response:

We thank the reviewer for this comment. We have added a figure showing the contributions of the excited state, the intermediate state, and the hot ground state to the Supplementary Information.

Authors' Action:

We have added the decomposition of a transient spectrum using the four-state kinetic model to the Supplementary Information, Supplementary Figure 7. We discuss the signatures of the excited state contributions (ground state bleach and excited-state absorption) and the similar spectral shape of the intermediate and the hot ground state spectra.

Comment 7:

What is the intermediate state – a librational excited water molecule?

Authors' Response:

This is an excellent question. We use the intermediate state in the kinetic model to describe the temporal dependence of the transient spectra to account for the delayed appearance of the spectral signatures of the hot ground state after vibrational relaxation of the bending mode excitation, as also reported in J. Chem. Phys. 147, 084503 (2017). As such, the intermediate state represents the state at which the bending mode energy has been redistributed to (more than one) lower frequency modes. Yet, in this intermediate state, the vibrational energy is not yet fully equilibrated as in the hot ground state. As such, the intermediate state subsumes the effect of the population of lower frequency modes, presumably including librations, but it can also contain other (including IR inactive) states.

Authors' Action:

We have added a more detailed discussion of the kinetic model to the Supplementary Note 8, which reads: “We use the intermediate state in the kinetic model to describe the delayed appearance of the spectral signatures of the hot ground state after vibrational relaxation of the bending mode excitation. This delayed appearance has also been reported in Ref. 18. The intermediate state is used as a means to model the initial re-distribution of the bending mode energy to lower frequency modes after de-population of the bending mode excitation. In the intermediate state, the bending mode is depopulated, but the excess vibrational energy is not yet fully equilibrated over all available modes. This equilibration is closer to the thermal equilibrium in the subsequently populated hot ground state, which – similar to the intermediate state – represents the population of a manifold of low energy states. Given that these lower frequency states are differently coupled to the experimentally interrogated bending mode, they give rise to different transient bending mode spectra.”

Comment 8:

Are the 0.28 ps data missing from figure 3a?

Authors' Response:

Thank you for pointing out this mistake. We have added the 0.28 ps data to Figure 3 (a) of the manuscript.

Comment 9:

To analyse the anisotropy decay has the Condon approximation been made, and is it appropriate?

Authors' Response:

Yes – this is a very good point. We did indeed make that approximation, and the validity of the Condon approximation was carefully checked in a previous simulation paper (J. Chem. Phys. 143, 014502 (2015)).

Authors' Action:

We mention in the Supplementary Information, Supplementary Note 9: “The anisotropy decay can be perturbed by the frequency-dependence of the transition dipole moment²⁵. This non-Condon effect is pronounced, for example, for the anisotropy decay of the O-H stretch mode. However, the non-Condon effects are negligible in the H-O-H bending mode²⁶. Thus, we analyzed the anisotropy decay of the H-O-H bending mode within the Condon approximation.”

Comments 10:

The discussion of the anisotropy data seems reasonable, but the 'faster than rotation' assignment needs some more discussion. The single molecule rotation time is 2 ps. Given the ca 200 fs population relaxation could pure rotation have been observed by this method.

Authors' Response:

Thank you for the comments. We agree with the reviewer that the fast population relaxation would not allow sampling the rotational motion over a full 2 ps time window. The anisotropy parameter is a measure of the orientational excitation memory and is normalized to the

isotropic (quickly decaying) signal. As such, rapid population relaxation limits the time window over which the orientational memory can be probed. If the excited state contributions can be accurately isolated from the transient data, the anisotropy dynamics will not be affected. Thus, if the loss of the orientational memory would be solely due to rotation of water with a rotation time of 2 ps, the anisotropy parameter would decay by a little over ~20% within our ~500 fs time window. In contrast, we find a ~70% decay within this window, which we explain by randomization of the transition dipoles via coupling to librations.

Authors' Action:

We have added this notion to the revised manuscript, which reads: “Consistent with these reports, and despite the short vibrational relaxation time limiting the accessible time window for determining the excitation anisotropy to $t < 600$ fs, we find the anisotropy to decay appreciably faster than what would be expected based on the known rotational dynamics of water, i.e., the H-O-H angular bisector orientational decay of 1.9 ps³⁵”.

Comments 11:

How is the anisotropy measurement influenced by the intermediate population?

Authors' Response:

The reviewer raises an important point regarding the effect of the intermediate state spectra (and also the hot ground state) spectra on the determined anisotropy decays. Typically, on short timescales, the initially dissipated energy is distributed locally around the excited oscillator, and thus the orientational memory of the excited oscillator is retained. Only at longer timescales, when the dissipated energy also modulates more distant chromophores, which were initially not excited, the contributions of relaxed states (states different than the excited bending mode) can give rise to a decay of the anisotropy (see e.g. J. Phys. Chem. A 2011, 115, 51, 14593–14598 and J. Phys. Chem. B 2015, 119, 4, 1558–1566). In our analysis, we consider two extreme cases: (i) All states, excited state, intermediate state, and hot ground state pertain the orientational excitation memory (i.e. the raw experimental anisotropies in Supplementary Figure 8) or (ii) the dissipated energy modulates all bending chromophores for the hot ground state and its contribution is isotropic and can be subtracted from the transient signals (as shown for the anisotropies in Figure 4 (a)). Both limiting cases show a rapid decay of the excitation anisotropy, faster than the rotational motion of water, which we explain by the rotational contributions to the bending mode.

Authors' Action:

We have added this discussion in detail to Supplementary Notes 9.

Comments 12:

Minor points

P11 4 lines from the bottom, 'and thirdly fluctuations' In what?

P12 4 lines from bottom, there is a misplaced comma.

Authors' Response:

We are sorry that our statement is not clear. Here the 'thirdly fluctuations' represents the orientational fluctuation caused by the bend-libration mixing (Figure 1(d)).

Authors' Action:

We revised the manuscript and fixed the mistakes.

Reviewer #2 (Remarks to the Author):

In this paper, the authors experimentally and through molecular dynamics simulations studied the vibrational coupling of the H-O-H bending mode of liquid water. The vibrational energy relaxation and orientational dynamics of H-O-H bending mode were also studied. The authors established that the intermolecular coupling of the bending mode is very weak. This fact has been known for a while (see e.g. J. Chem. Phys. 143, 014502 (2015)). Importantly, the authors found that there is a strong coupling of the bending mode and librations which gives rise to the 1650 cm^{-1} band and demonstrated that the librational mode plays a key role in the energy transfer pathway and orientational dynamic of the bending mode band. This paper should be of considerable interest to researchers studying condensed-phase water. Publication in Nat. Comm. is recommended. I have only two minor suggestions.

Page 3: "The O-H stretch vibration in liquid water is quasi-instantaneously delocalized across several water molecules, and is strongly coupled to the bending mode". Considering a substantial difference in vibrational frequencies it is not clear why the authors say that these modes are strongly coupled. The authors should be more specific about what they mean by coupling here.

Page 6. "Furthermore, the degree of narrowing is much smaller than expected from previous theoretical predictions [29]". Ref. 29 illustrates a very good agreement between theory and experiment, yet the authors of this paper say that the narrowing is smaller. How can these two facts be reconciled? In other words, why the experimental data presented in this paper appear to be different from the experimental data used as a reference in Ref. 29?

Authors' Response:

Thank you for the detailed comments on our work. In the following, we respond to the reviewer's comments.

Comment 1:

Page 3: "The O-H stretch vibration in liquid water is quasi-instantaneously delocalized across several water molecules, and is strongly coupled to the bending mode". Considering a substantial difference in vibrational frequencies it is not clear why the authors say that these modes are strongly coupled. The authors should be more specific about what they mean by coupling here.

Authors' Response:

We thank the reviewer for his comments and apologize for the ambiguity. According to Ref. J. Chem. Phys. 147, 084503 (2017), the source of the coupling of the bending mode and stretch mode is the anharmonic interaction of the stretch mode and bending mode.

Authors' Action:

We have revised the statement accordingly and added the above-mentioned reference: "Within one molecule, coupling can occur between the stretch modes of the two O-H groups⁸, and strong intermolecular coupling of stretch modes makes its excitation quasi-instantaneously delocalized across several molecules in liquid water. The stretch modes are further coupled to the bending mode and its overtone via anharmonic interactions⁹⁻¹⁷,"

Comment 2:

Page 6. "Furthermore, the degree of narrowing is much smaller than expected from previous theoretical predictions [29]". Ref. 29 illustrates a very good agreement between theory and experiment, yet the authors of this paper say that the narrowing is smaller. How can these two facts be reconciled? In other words, why the experimental data presented in this paper appear to be different from the experimental data used as a reference in Ref. 29?

Authors' Response:

In Ref. 29, the authors took two different data set (pure H₂O data from S. Ashihara, S. Fujioka, and K. Shibuya, Chem. Phys. Lett. 502, 57 (2011), and the isotopically diluted water data from L. Piatkowski and H. J. Bakker, J. Chem. Phys. 135, 214509 (2011).) The exact reason why the authors of ref. 29 arrive at a different conclusion, remains unclear to us; it may well be related to different procedures to extract the background absorptions from the two different data sets. To avoid such potential bias, we have performed three different spectroscopies, and explained the detail of the background subtraction procedure in the Supplementary Note 6.

For our data, we find that the background subtraction can reduce the FWHM (See the Supplementary Figure 4) and the thus obtained 95 cm⁻¹ FWHM, is in good agreement with the ~90 cm⁻¹ reported in Chem. Phys. Lett. 502, 57 (2011).

Authors' Action:

To demonstrate the effect of the background subtraction we have added the raw and corrected IR spectra to the Supplementary Information (Supplementary Figure 4).

Reviewer #3 (Remarks to the Author):

This manuscript describes a combined experimental and theoretical investigation aimed at characterizing the inter and intramolecular couplings and energy dissipation pathways of the water bending mode. Where stretch mode have been extensively studied, the bending mode is understudied despite it being essential to the energy dissipation of water. The study explains the much different inter vs intra-molecular contributions to the mode coupling of the bend vs the stretch and further show that the bending mode is heavily influences by libration modes giving rise to a much different mode character. This is a very interesting study that is well-carried out and described. It will be interesting to a broad audience and appropriate for Nature Communications.

My only comments pertain to a couple of wording choices that needs to be corrected before publication:

Comment 1:

Page 3: "Within one molecule, coupling can occur between the symmetric and antisymmetric O-H stretch modes⁸," The paper from the Tokmakoff Group (Reference 8) show that the mode character is dramatically different between liquid D₂O and H₂O. In liquid D₂O there remain some character of the gas phase symmetric and asymmetric stretch modes but this is not the case for liquid H₂O. The authors statement is thus misleading since this is only partly true for D₂O and not for H₂O. The statement should be reformulated so something like: "Within one molecule, coupling can occur between the two OH stretch modes"

Authors' Response:

We are grateful to the reviewer for pointing this out, and agree with the reviewer's comment.

Authors' Action:

We have revised the statement in the manuscript: "Within one molecule, coupling can occur between the stretch modes of the two O-H groups⁸, and strong intermolecular coupling of stretch modes makes its excitation quasi-instantaneously delocalized across several molecules in liquid water."

Comment 2:

Page 9: "the excited state population relaxes to the ground state, and the dissipated energy gives rise to a temperature increase, which results in persistent spectral modulations at long delay times." This is partly semantics, but at the probed timescales the dissipated energy among low-frequency modes cannot be described by a thermal distribution and thus cannot be described as a temperature increase. Only at much longer timescales than is probed in this

experiment can this be described as a temperature increase. The statement should be reworded accordingly.

Authors' Response:

We agree with the reviewer that a temperature rise may be not adequate for describing the transient signals at delay times as short as 0.5 ps.

Authors' Action:

We revised the manuscript accordingly: "With increasing delay time, the excited state population relaxes to the ground state, and the dissipated energy populates low frequency modes, which results in persistent spectral modulations at long delay times."

REVIEWERS' COMMENTS

Reviewer #1 (Remarks to the Author):

All on my comments have been addressed in a satisfactory way. The only other point to raise is on presentation. The axes label 'intensity' is probably not right for FTIR (and probably should not be negative in Supplementary Fig 4).

Similarly the label for figure 4 should probably be anisotropy rather than anisotropy decay.

Aside from that the paper is suitable for NCOMM and may be published.

Reviewer #2 (Remarks to the Author):

I am satisfied with the authors' reply to my comments and I recommend this manuscript for publication in Nat. Comm. after a few small issues below are addressed:

Page 6 lines 101-103: "The insensitivity of the

magnitude of the shift to the experimental details is consistent with the notion that coupling of the bending mode is not of dipolar nature but has mechanic or electronic origins". Coupling to what? Do the authors imply the intermolecular bending-bending mode coupling here?

Page 17 lines 256-257: "The weak mode mixing for the stretch mode..." Do the authors refer to isotopically diluted water here as well? If so it should be stated explicitly. If not, then, the corresponding coupling is actually strong, as the authors acknowledged before (linear 50-51 of page 3).

Reviewer #3 (Remarks to the Author):

I am satisfied with the authors responses and will recommend the manuscript published as is.

REVIEWERS' COMMENTS

Reviewer #1 (Remarks to the Author):

All on my comments have been addressed in a satisfactory way. The only other point to raise is on presentation. The axes label 'intensity' is probably not right for FTIR (and probably should not be negative in Supplementary Fig 4).

Similarly the label for figure 4 should probably be anisotropy rather than anisotropy decay.

Aside from that the paper is suitable for NCOMM and may be published.

Comment 1:

The axes label 'intensity' is probably not right for FTIR (and probably should not be negative in Supplementary Fig 4).

Authors' Action:

We appreciate the reviewer's comments and have modified the y-axis label (From 'Intensity' to 'Absorbance') for FTIR in Figure 2 of the manuscript and Supplementary Fig 4. Besides, we replot Supplementary Fig 4 and now it doesn't show negative in it.

Comment 2:

Similarly the label for figure 4 should probably be anisotropy rather than anisotropy decay.

Authors' Action:

We agree with the reviewer and modified the y-axis label (From 'Anisotropy decay' to 'Anisotropy') in Figure 4.

Reviewer #2 (Remarks to the Author):

I am satisfied with the authors' reply to my comments and I recommend this manuscript for publication in Nat. Comm. after a few small issues below are addressed:

Page 6 lines 101-103: "The insensitivity of the magnitude of the shift to the experimental details is consistent with the notion that coupling of the bending mode is not of dipolar nature but has mechanic or electronic origins". Coupling to what? Do the authors imply the intermolecular bending-bending mode coupling here?

Page 17 lines 256-257: "The weak mode mixing for the stretch mode..." Do the authors refer to isotopically diluted water here as well? If so it should be stated explicitly. If not, then, the corresponding coupling is actually strong, as the authors acknowledged before (linear 50-51 of page 3).

Comment 1:

Page 6 lines 101-103: "The insensitivity of the magnitude of the shift to the experimental details is consistent with the notion that coupling of the bending mode is not of dipolar nature but has mechanic or electronic origins". Coupling to what? Do the authors imply the intermolecular bending-bending mode coupling here?

Authors' Response:

We apologize for this ambiguous statement. Here, the coupling means the intermolecular bending-bending mode coupling

Authors' Action:

We have revised the manuscript, which now reads "The insensitivity of the magnitude of the shift to the experimental details is consistent with the notion that intermolecular bend-bend coupling is not of dipolar nature but has mechanic or electronic origins³¹."

Comment 2:

Page 17 lines 256-257: "The weak mode mixing for the stretch mode..." Do the authors refer to isotopically diluted water here as well? If so it should be stated explicitly. If not, then, the corresponding coupling is actually strong, as the authors acknowledged before (linear 50-51 of page 3).

Authors' Action:

We thank the reviewer for this comment and apologize for the omission of this detail. We have revised the manuscript, which now reads "In contrast, the weak mode mixing for the stretch mode of isotopically diluted water is consistent with the observation that the anisotropy decay."

Reviewer #3 (Remarks to the Author):

I am satisfied with the authors responses and will recommend the manuscript published as is